☀️ PLOS | ONE

# Use of IoT sensing and occupant surveys for determining the resilience of buildings to forest fire generated PM$_{2.5}$

**Jovan Pantelic**[1]*, **Megan Dawe**[1], **Dusan Licina**[2]

**1** Center for the Built Environment, University of California, Berkeley, California, United States of America,
**2** Human-Oriented Built Environment Lab, School of Architecture, Civil and Environmental Engineering,
École Polytechnique Fédérale de Lausanne, Lausanne, Switzerland

\* pantelic@berkeley.edu

**Data Availability Statement:** All relevant data are within the paper and its Supporting Information files.

**Funding:** The authors would like to acknowledge CITRIS at University of California Berkeley for

## Abstract

Wildfires and associated emissions of particulate matter pose significant environmental and health concerns. In this study we propose tools to evaluate building resilience to extreme episodes of outdoor particulate matter using a combination of indoor and outdoor IoT measurements, coupled with survey-based information of occupants' perception and behaviour. We demonstrated the application of the tools on two buildings with different modes of ventilation during the Chico Camp fire event. We characterized the resilience of the buildings on different temporal and spatial scales using the well-established I/O ratio and a newly proposed E-index that evaluates indoor concentration in the context of adopted 24-hour exposure thresholds. Indoor PM$_{2.5}$ concentration during the entire Chico Camp Fire event was 21 µg/m$^3$ for 4$^{th}$ Street (Mechanically Ventilated) and 36 µg/m$^3$ for Wurster Hall (Naturally Ventilated). The cumulative median I/O ratio during the fire event was 0.27 for 4$^{th}$ Street and 0.67 for Wurster Hall. Overall E-index for 4$^{th}$ Street was 0.82, suggesting that the whole building was resilient to outdoor air pollution while overall E-index was 1.69 for Wurster Hall suggesting that interventions are necessary. The survey revealed that occupant perception of workplace air quality aligns with measured PM$_{2.5}$ in the two buildings. The results also highlight that a large portion of occupants wore face masks, even though the PM$_{2.5}$ concentration was below WHO threshold level. The results of our study demonstrate the utility of the proposed IoT-enabled and survey tools to assess the degree of protection from air pollution of outdoor origin for a single building or across a portfolio of buildings. The proposed survey tool also provides direct links between the PM$_{2.5}$ levels and occupants' perception and behavior.

## 1. Introduction

In recent years, we have observed increased wildfire frequency and intensity in response to global warming [1]. Many regions prone to fire are forecasted to have increased frequency of wildfires and associated air pollution episodes [2]. Wildfire smoke includes carbon dioxide,

sponsoring this research with the grant number 69085-24003-44-NHJVN. This research was also partially supported by the Republic of Singapore's National Research Foundation through a grant to the Berkeley Education Alliance for Research in Singapore (BEARS) for the Singapore-Berkeley Building Efficiency and Sustainability in the Tropics (SinBerBEST) Program.

**Competing interests:** The authors have declared that no competing interests exist.

water vapor, carbon monoxide, particulate matter (PM), complex hydrocarbons, nitrogen oxides, trace minerals, and several thousand other compounds [3]. There is clear evidence that wildfire smoke, which includes particulate matter with 2.5 μm diameter ($PM_{2.5}$), is linked to respiratory health implications, morbidity, and mortality [4,5]. In the U.S., mathematical models show that premature deaths attributed to wildfire generated $PM_{2.5}$ exposure will double compared to the early 21st century [6]. Furthermore, studies have shown that wildfire smoke exposure during pregnancy affected birth weight among term infants [7], and that mental health symptoms increased among an adolescent population exposed to wildfire smoke [8]. In planning for the future, cities, communities, and building ecosystems in wildfire prone regions need to be resilient to the associated wildfire effects [9], and provide safe indoor air quality (IAQ) to protect occupants.

In order to achieve appropriate IAQ, the building industry must develop standards for protecting building occupants against wildfire air pollution, which do not currently exist [10,11]. Building ventilation standards such as ASHRAE 62.1 and 62.2 only deal with typical outdoor air conditions (i.e., not under extreme air pollution events). The available guidelines include the British Columbia Center for Disease Control Wildfire Smoke Response Planning document that states: *"More than one portable air cleaning unit may be required for large rooms or homes with high air change rates"*. WHO/UNEP/WMO Guidelines for Vegetation fire events provide general recommendations, such as "reduce infiltration" or install and maintain "effective filters" [12]. The California Department of Health [13] provide similar guidance. WHO and EPA provide short-term (24-hr average) and long-term (annual) $PM_{2.5}$ exposure thresholds, above which there are long-term human health impacts. The existing guidelines are largely qualitative without quantifiable instructions, such as an appropriate filter grade, air pollution monitoring and ventilation operation. Few studies have examined personal particle exposure indoors during air pollution events [14,15]. A study by [16] is one of the few that examined infiltration of wildfire generated air pollution into buildings. A major barrier to establishing standards and protecting occupants is that very few (if any) buildings have equipment to measure indoor PM or pollutants, besides $CO_2$ and CO.

Internet of Things (IoT) environmental sensing platforms can be placed indoors and outdoors to measure diverse air pollutants, including $PM_{2.5}$. IoT $PM_{2.5}$ measurements can be used to understand the relationship between outdoor and indoor $PM_{2.5}$ levels and pathways by which outdoor particles enter indoors. These relationships can be described with known metrics such as Indoor to Outdoor (I/O) ratio, infiltration factor ($F_{in}$), and penetration factor (P) [17]. The studies have concluded that low-cost IoT $PM_{2.5}$ sensors have reasonable accuracy for building scale implementation [18,19,20] and that they are sufficiently accurate for identifying high air pollutant concentrations, as is observed in wildfires [21].

The effect of wildfire generated air pollution on human health and mortality highlight the importance of managing exposure. However, the barriers to reducing indoor exposure are three-fold. First, there are no quantifiable guidelines or standards in place. Second, there is a lack of measurement equipment in buildings for wildfire generated air pollutants, such as $PM_{2.5}$. Lastly, without measurement equipment, the building industry is left with only the existing qualitative guidelines without the ability to quantify the effectiveness of resiliency strategies. This all leads to a significant gap in defining and quantifying building resiliency, effectiveness of chosen interventions, and general understanding of occupant exposure indoors. The objective of this study was to address the second two barriers by demonstrating how a combination of an IoT environmental sensing network implemented locally outdoors and inside the building and an occupant survey can quantify building resiliency to urban scale air pollution. In doing this, we applied known analytical tools and developed another. Specifically, we analyzed $PM_{2.5}$ measurements during a wildfire event at different spatial (whole

building or room scale) and temporal scales (instantaneous and long term) to demonstrate the application of building assessment tools. We used two buildings located in Berkeley (California) during the two-week Chico Camp fire in November 2018; one building was equipped with mechanical ventilation and the other with natural ventilation. By reliance on combination of I/O ratio metric, survey questionnaires and a newly proposed E-index, the study sought to quantify the resilience of buildings against particle penetration indoors from outdoor origin.

## 2. Materials and methods

We used two types of indoor $PM_{2.5}$ sensors (Clarity Inc, and Senseware), one type of outdoor $PM_{2.5}$ sensor (Clarity Inc), and one type of $CO_2$ sensor (Senseware) placed on and in two commercial buildings in Berkeley, CA. One of the buildings was naturally ventilated, while the other was mechanically ventilated. We also developed and distributed a survey to gather feedback on occupant experience in their workplace during the Chico Camp fire compared to their typical experience in the space. With this data, we developed and evaluated tools to quantify and assess each building's resilience to the extreme air pollution episode.

### 2.1 Experimental apparatus

Both Clarity and Senseware $PM_{2.5}$ nodes (Fig 1) count the particle number using the principle of light scattering, which are characteristic for optical particle counters. The accuracy of the all the sensors was the same—within ±10 μg/m$^3$ in the range of 0 to 100 μg/m$^3$ and ±10% in the range of 100 to 1000 μg/m$^3$. Data was collected on 15-minute intervals for Clarity nodes on 1-minute intervals for Senseware nodes. As typical for optical particle counters, Clarity and Senseware nodes are factory-calibrated with Arizona Test Dust (ATD). Calibration with ATD does not robustly account for many aerosol particle properties, such as density, shape, refractive index, and absorption. Therefore, if aerosol properties of the particles measured in the field do not have properties similar to ATD (as is likely the case for wildfires), the optical particle counters can produce inaccurate readings. There is a lack of understanding of the wildfire source profiles and resulting aerosol composition [22]; thus making the field calibration challenging. To arrive at accurate measurements, the Clarity nodes apply co-location of nodes and post-processing correction factors to the raw outdoor measurements using a local government measurement stations that indirectly include information on particle optical properties. For this dataset, the Clarity node corrections for particle optical properties were extracted from the California Air Resources Board Oakland Laney College measurement station. Clarity Inc. typically only applies these corrections to outdoor nodes; however, we applied

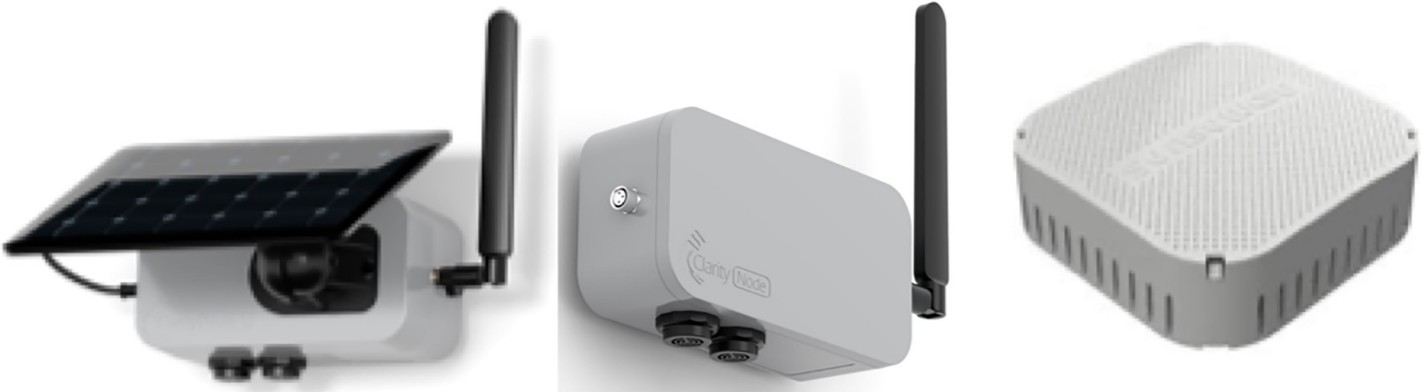

**Fig 1.** IoT sensors used in the study: Clarity outdoor node (right); Clarity indoor note (middle); Senseware indoor node (left).

the same correction to indoor Clarity nodes because under the Chico Camp fire scenario indoor $PM_{2.5}$ dominantly originated from the outdoors. Senseware nodes do not make corrections to the raw measurements and are calibrated with the ATD.

Senseware $CO_2$ sensors measured $CO_2$ levels with an accuracy of ±50 ppm. In Wurster Hall, the $CO_2$ sensors were paired with $PM_{2.5}$ measurements, while in 4[th] Street they were placed in two representative locations. Senseware contact monitors—(Model COZIR-LP) were mounted on the windows in Wurster Hall to detect window position. Detection was binary (i.e., open/closed) and did not provide information on opening area or window angle.

## 2.2 Quality assurance

Prior to deployment, Clarity Inc. and Senseware $PM_{2.5}$ nodes were compared side-by-side and with Grimm aerosol spectrometer (model 11-A, GRIMM Aerosol Technik GmbH, Ainring, Germany) that provides time-and size-resolved data for aerosol particles. Clarity and Senseware $PM_{2.5}$ nodes were within ±8 μg/m$^3$ compared to Grimm measurements at ambient $PM_{2.5}$ levels ranging from 10 μg/m$^3$ to 45 μg/m$^3$, which is within the given accuracy for each node.

## 2.3 Experimental design

We selected two buildings, 1608 4th Street Building (mechanically ventilated) and Wurster Hall (mixed-mode ventilation operating as fully naturally ventilated during the study period) to deploy sensors. Both buildings were located in Berkeley (California). In March 2018, we deployed indoor and outdoor $PM_{2.5}$ sensors, indoor $CO_2$ sensors in both buildings and window contact sensors in Wurster Hall. Berkeley was affected by air pollution from the Chico Camp fire from November 8[th] until November 21[st], 2018. The current study provides analysis of data collected during the Chico Camp fire.

*1608 4th Street Building (abbreviated as 4[th] Street)* is a mechanically ventilated building with centralized heating, ventilation and air-conditioning (HVAC) system. The HVAC system utilizes three stages of air filtration: the first stage of filtration is by a MERV 8 pleated filter, the second stage is the Gas Phase filter, and the final filter stage is a high-efficiency MERV 13 filter. The measure $CO_2$ decay profile during non-occupied hours were used to calculate infiltration rates (details in the Supporting Information). Infiltration was below 0.3 air exchanges per hour suggesting that the building is airtight without operable windows. The building has approximately 270 full-time occupants.

*Wurster Hall*—Mixed-mode building with seasonal changeover, meaning that the building is fully naturally ventilated when the building is operating in cooling, as was the case during the Chico Camp fire. The building relies on operable windows for ventilation and cooling. The building has a high level of infiltration, as shown by typical $CO_2$ levels below 550 ppm during normal operation. The building has approximately 300 full-time occupants.

**2.3.1 Placement of outdoor and indoor sensors.** *1608 4th Street Building*—is occupied on the 2[nd], 3[rd] and 4[th] floor. Floors had the same layouts; therefore, considering the available number of sensors, we placed all sensors on the 3[rd] floor. We placed one outdoor $PM_{2.5}$ sensor (Clarity node) on the roof of the building, and 15 $PM_{2.5}$ sensors (14 Senseware nodes and one Clarity node) at different indoor locations. S1 Fig in the Supporting Information provides a floor plan with sensor locations in the 4[th] Street. Each of the floors consists of large open floor office area with high partitions housing 71 stations, six private offices, two team meeting rooms, one large team meeting room, and two phone booths. We placed one $PM_{2.5}$ and one $CO_2$ sensor in the private office, and the remaining sensors in the open floor office. We wanted to measure if the building envelope was leaky, so we placed $PM_{2.5}$ sensors near outside walls with windows so that we can conservatively estimate building operation.

*Wurster Hall*—consisted of two building wings with separate HVAC systems. The south wing is four stories and consisted of classrooms, open plan offices, small and large meeting rooms, and several private offices. The north wing is 10 stories and consisted almost entirely of large open plan offices. Our study focused on the south wing of the building. We placed one $PM_{2.5}$ sensor (Clarity node) on the roof above the 4[th] floor, and 11 $PM_{2.5}$ sensors (10 Senseware nodes and one Clarity node). Private offices were the most common space type in Wurster Hall's south wing; therefore, we placed the majority of $PM_{2.5}$ and $CO_2$ sensors in that environment. Additionally, we placed sensors in one meeting room with operable windows and in one small open plan office with four occupants. This particular open plan office was chosen also because it was adjacent to the hallway with an entrance door to the building. S2 Fig in the Supporting Information provides a floor plan with sensor locations in the Wurster Hall. We placed contact sensors on operable windows; each room had at least one manually operated awning-type window, and some rooms had two or more depending on the size of the space.

## 2.4 Data analysis

An assumption for data analysis was that the dominant source of indoor $PM_{2.5}$ originated from the outdoors. Although this assumption is not generally valid at typical outdoor $PM_{2.5}$ concentrations, in this case, the $PM_{2.5}$ data collected before and after the Chico Camp Fire showed typical indoor $PM_{2.5}$ concentrations below 3 μg/m$^3$, implying that indoor sources were not significant relative to the outdoor $PM_{2.5}$ during forest fires. The average indoor raw $PM_{2.5}$ concentrations (i.e., not corrected for particle properties) measured in each building during the Chico Camp fire followed trends of outdoor measurements and were approximately 80 times greater than the average concentrations measured the week before and after the Chico Camp fire. This validates the assumption that all indoor particles can be treated as having outdoor origin during the Chico Camp fire.

As mentioned, Clarity Inc. corrects measured data to include information about $PM_{2.5}$ composition, while Senseware measurements rely on ATD calibration. Due to this disparity, the analysis uses raw measurements when combining or comparing data from the Clarity and Senseware nodes, and the Clarity corrected measurements when discussing observed indoor and outdoor $PM_{2.5}$ concentrations. For the analysis, we define building operating hours as 07:00 to 19:00. When presenting the results, we identify the median value with the interquartile range in parentheses (i.e., the range between the 25[th] and 75[th] percentiles), for example median $PM_{2.5}$ concentration of 21 μg/m$^3$ (IQR = 13 μg/m$^3$).

**2.4.1 Outdoor $PM_{2.5}$ comparison.** We compared local (i.e., building) outdoor $PM_{2.5}$ concentrations to nearby regional weather station measurements to evaluate the current assumption that regional measurements are representative of individual exposure.

**2.4.2 Building assessment tools.** We assessed the following tools for characterizing building resiliency during extreme pollution events based on IoT environmental sensing:

(i). Comparison of indoor $PM_{2.5}$ concentrations to the WHO exposure threshold value for $PM_{2.5}$. We chose the WHO guidelines because it is the strictest, with the maximum 24-hour mean exposure of 25 μg/m$^3$. Alternative guidelines such as EPA may be used, depending on the region of the world. We compared median hourly indoor $PM_{2.5}$ concentration (i.e., median value from all indoor sensors) to WHO 24-hour mean exposure guideline, and calculated the Exceedance index (E-index), as shown in Eq 1, on an hourly basis. The E-index, is a unitless value that informs by how much hourly $PM_{2.5}$ concentration exceeds the recommended level. We can then calculate the number or percent of hours that indoor $PM_{2.5}$ concentration exceeded specified levels during the air pollution

episode, or the whole building average E-index. It is a tool to evaluate occupant exposure that can be compared across buildings during extreme air pollution events. This can also be done on a space-by-space basis within the same building.

$$E = \frac{Cmeasured\ PM_{2.5}}{25\ \mu g/m^3}$$

Eq (1)

(ii). Comparison of indoor and outdoor PM$_{2.5}$ concentrations. Outdoor PM$_{2.5}$ can enter the indoor environment mainly by penetration (uncontrolled via building cracks and doors) or ingress (controlled via mechanical ventilation system or windows). From the building operation perspective, quantification of pathways and the quantity of outdoor particles that enter the indoor environment is one of the most important aspects of building resilience to extreme air pollution events. Considering negligible indoor PM$_{2.5}$ sources, we quantified PM$_{2.5}$ penetration using indoor to outdoor particle (*I/O*) ratio (Eq 2) using outdoor and indoor PM$_{2.5}$ measurements. Eq 2 is appropriate when the indoor pollution sources can be neglected, as we determined is the case for this event. We can calculate I/O ratio at different temporal (i.e., cumulative episode or hourly) and spatial (i.e., whole building or by room) scales, which informs different scenarios of interest. The differences between I/O ratios among nearby buildings is typically attributed to differences in indoor particle sources, geometry of building cracks, wind direction and intensity, ventilation strategy and rate, and air filtration (16). We supplemented the I/O ratio analysis with information on infiltration rates calculated using $CO_2$ decay and knowledge of the HVAC system operation.

$$I/O = \frac{C_{in}(t)}{C_{out}(t)}$$

Eq (2)

*Instantaneous I/O ratio* was calculated for each indoor sensor location using hourly mean indoor and outdoor PM$_{2.5}$ concentration (Eq 2, where *C$_{in}$(t)* and *C$_{out}$(t)* are the hourly means). We defined instantaneous as hourly in this study, but the calculation could be done at smaller time steps if necessary for a building or space. We used raw measurements from all sensors so data was comparable between sensor manufacturer. To calculate the whole building *Instantaneous I/O ratio*, we used the median hourly mean PM$_{2.5}$ from the indoor sensors compared to the hourly mean outdoor PM$_{2.5}$ concentration; we used the median value instead of the mean because it is robust to outlier instances, such as a window being left open which occurred in one space. *Instantaneous I/O ratio* is a tool that enables quick-response comparison between buildings and between spaces within a building on a short time scale.

*Cumulative I/O ratio* was calculated for each building to represent the overall whole building I/O ratio over a given period of time post-event, which was the Chico Camp fire period for this study. It is the median value of all the mean hourly I/O ratios at each sensor location in a building during an event. While the *Instantaneous I/O ratios* can vary depending on environmental conditions and driving forces (e.g. air temperatures and outdoor wind speeds), the *Cumulative I/O ratio* provides an overall value to compare buildings' abilities to prevent infiltration and penetration of outdoor pollutants during extreme pollution events. The *Cumulative I/O ratio* can also be calculated at each sensor location to compare between spaces within a building.

**2.4.3 Survey of occupants.** It may not always be feasible to measure indoor and outdoor PM$_{2.5}$ for every building due to sensor availability and cost. A survey is an economical and

simple method to gather data; however, it relies on self-reported information from occupants which may not always reflect the environmental conditions. This study compares the indoor $PM_{2.5}$ measurements with occupant feedback on perceived air quality, impacts to productivity from air quality, and behavior changes during the Chico Camp fire compared to their typical experience in the space. We then evaluated the potential effectiveness of surveys as a building assessment tool.

The voluntary web-based survey was distributed to full-time occupants in each building between March 13 and March 22, 2019. University of California at Berkeley Institutional Review Board specifically approved this study (obtained Protocol ID:2019-02-11802). Although there is a four-month gap between the fire and occupants' survey responses, the intensity of air pollution during the Chico Camp fire was atypical enough for the Berkeley area that we think the responses are valid representations of occupants' perception and behavior during that time. The survey used a five-point scale from -2 to +2 to rate occupant satisfaction from "dissatisfied" to "satisfied" or"interfered" to"enhanced", with a neutral vote as the middle value. The survey had two parts: first, the occupants were asked about their typical experience in the space, and then secondly, about their experiences during the Chico Camp fire. The survey main topics and rating scale included items listed in Table 1, and survey questions can be found in Supplemental information (Wildfire air quality survey).

Campus closures occurred on Nov 16, 19, and 20 in response to outdoor pollutant levels, with the exception of some services that included staff in 4[th] Street; therefore, many 4[th] Street occupants worked in their building during the closure while Wurster Hall was closed. The survey controlled for and separated out these days to understand whether occupants in either building worked in their workplace on those days.

## 2.5 Statistical tools used

We analyzed the IoT sensing and survey data using R version 3.5.0 software [23]. We performed statistical analysis on the measured sensor data to compare between sensor locations (i.e., between buildings, between buildings and a regional weather station, and within buildings) and to compare survey responses between buildings and between typical conditions (i.e., non-forest fire) and forest fire responses. The data under consideration was not normally distributed, so we used non-parametric tests. To assess statistical significance between measured $PM_{2.5}$ concentrations at different locations, we used a two-sided Wilcoxon Rank-Sum test, also known as Mann-Whitney test. To compare IAQ satisfaction survey responses between buildings, we also used a two-sided Wilcoxon Rank-Sum test. We could have used a one-sided test because the hypothesis is that occupants in Wurster Hall had lower satisfaction than those in 4th Street, but we used a more conservative test. We tested all scenarios to a 95% confidence

**Table 1. Surveyed topics and response scales provided to survey respondents.**

| Survey topic | Five-point scale values -2 -1 0 1 2 |
|---|---|
| Satisfaction with indoor air quality | Dissatisfied, slightly dissatisfied, neutral, slightly satisfied, satisfied |
| Impact of indoor air quality on work productivity | Interfered, slightly interfered, neither interfered/enhanced, slightly enhanced, enhanced |
| Self-reported health impacts[(a)] | Never, sometimes, often |
| Behavior and protective devices used during the Chico Camp fire | Never, rarely, sometimes (1/wk), often (2-3/wk), daily |

[(a)] Only provided three response options instead of five

level, meaning statistical significance was considered when the p value was below 0.05. There is no guidance for effect size specific to occupant survey responses; we used Cohen's d value and effect size recommended thresholds from [24] and [25].

## 3. Results and discussion

### 3.1 Outdoor $PM_{2.5}$ results: Spatial variability

Results in Fig 2 shows (a) hourly mean and (b) cumulative outdoor $PM_{2.5}$ concentration during the Chico Camp Fire for the two study buildings and four CARB $PM_{2.5}$ measurement stations located within 24 km from the study buildings: Aquatic Park, Laney College, West Oakland and Oakland International Blvd. The results show that concentration peaks and median mass concentrations for 4th Street and Wurster Hall are different. Analysis of the results in Fig 2B shows that median outdoor levels between the two study sites differ by 17 μg/m$^3$, even though the two locations were ~3.5 km away. Based on Wilcoxon Signed-Rank test, the difference between mean outdoor $PM_{2.5}$ measurements for the two buildings is statistically significant ($p<0.05$), the difference in means between Wurster Hall and Laney College station is statistically significant ($p<0.05$), but the difference in means between 4th Street and Laney College station is not ($p = 0.11$). Laney College station was chosen as a reference because it was the closest weather station and was used in the Clarity Inc. sensor calibration. Results point out the importance of local sensing, and that any method to evaluate building resilience with respect to the outdoor $PM_{2.5}$ levels needs to use local measurements.

### 3.2 Indoor $PM_{2.5}$ results

Results in this section include (i) indoor $PM_{2.5}$ concentrations and (ii) I/O ratio. In order to appropriately inform conclusions about the buildings from I/O ratios, we referenced information about the building infiltration and HVAC system operating hours to understand the underlying mechanisms that contributed to particle penetration and ingress.

**3.2.1 Comparison of $PM_{2.5}$ concentration to WHO guidance.** We evaluated measured indoor $PM_{2.5}$ concentrations to assess building resilience from the occupant exposure perspective because it affects occupant health, productivity, and comfort. Results in Fig 3 show that the median hourly indoor $PM_{2.5}$ concentration over the entire Chico Camp Fire event was

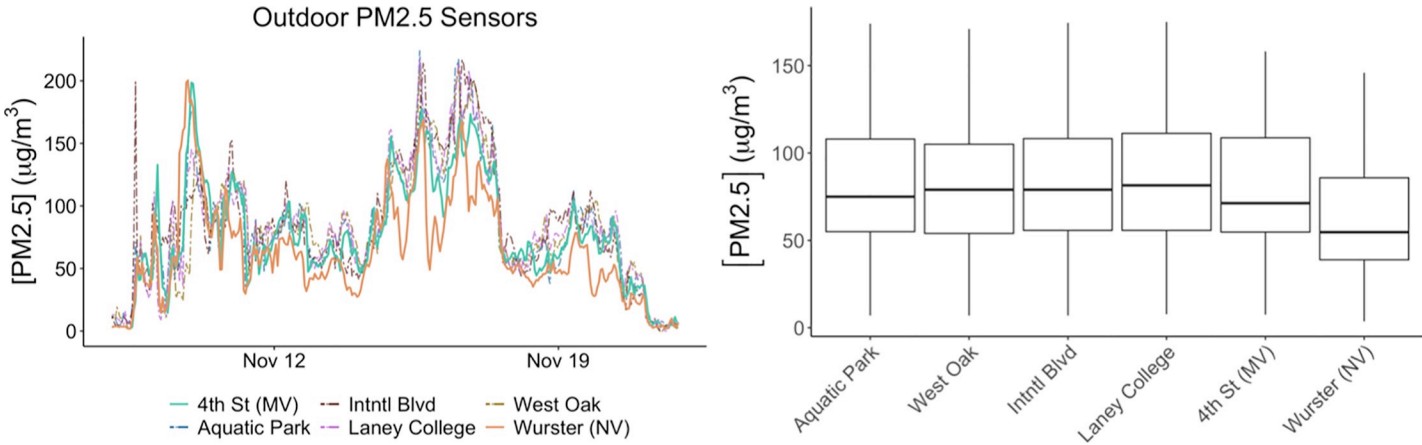

**Fig 2.** Hourly averaged outdoor PM measurements for the two study sites, and CARB aquatic park (closes to both sites), Laney College, West Oakland and International Blvd: a) hourly $PM_{2.5}$ concentrations; b) cumulative box plot that includes the 25th percentile, median, and 75th percentile, and the whiskers represent the 5th and 95th percentiles.

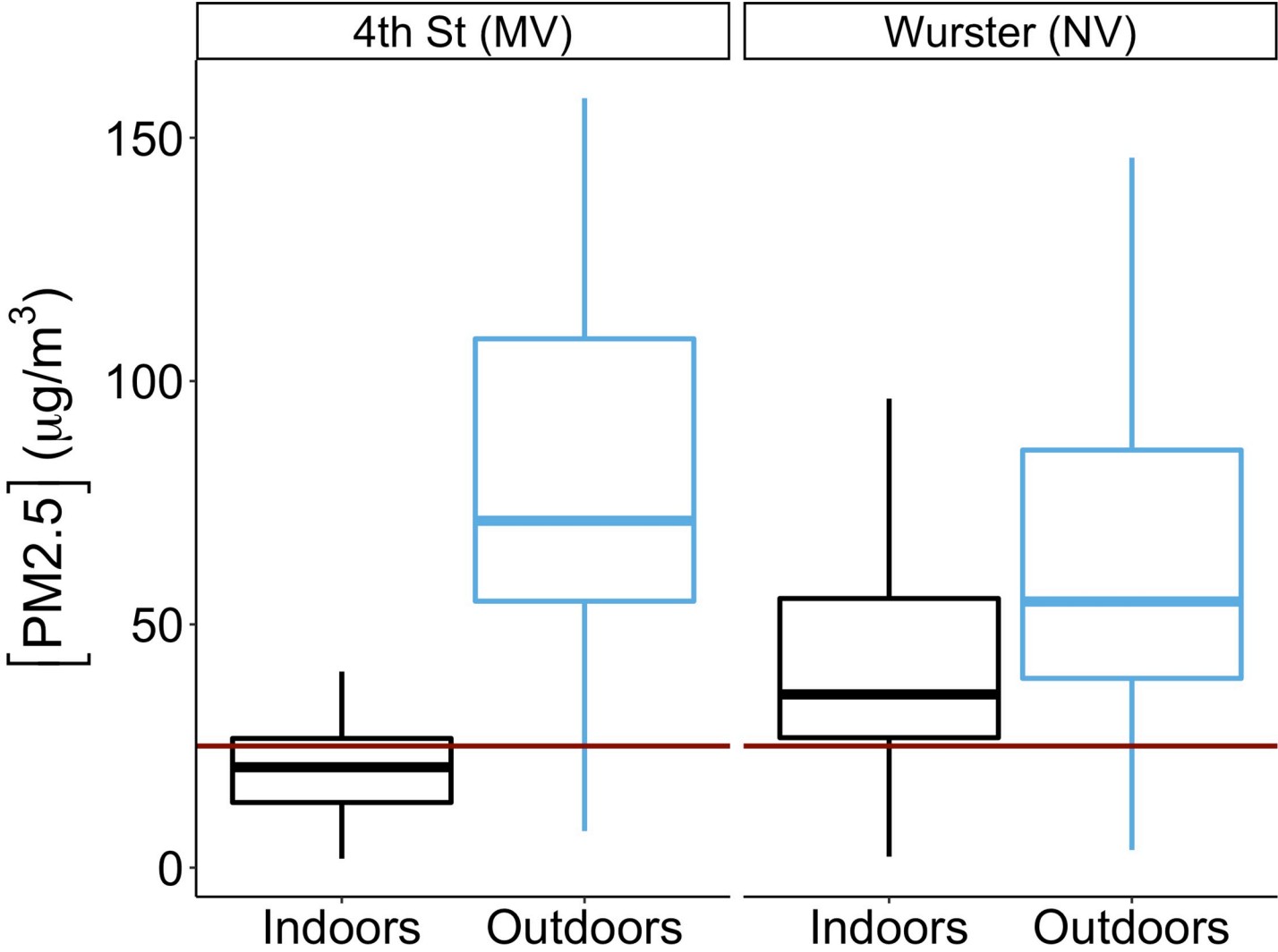

**Fig 3. Comparison of hourly PM₂.₅ concentrations between the two sites for the entire period of air pollution episode.** The box plot represents the 25th percentile, median, and 75th percentile, and the whiskers represent the 5th and 95th percentiles.

21 $\mu g/m^3$ (IQR = 13 $\mu g/m^3$) for 4th Street and 36 $\mu g/m^3$ (IQR = 29 $\mu g/m^3$) for Wurster Hall. Theses values represent a general overview of the whole building indoor levels, which indicates that there was a difference in exposure levels between the two buildings. The overall whole building indoor PM₂.₅ levels provide a coarse indicator of IAQ when they are benchmarked against recommended thresholds, such as WHO 24 hr exposure limit of 25 $\mu g/m^3$ or EPA limit of 35 $\mu g/m^3$. Based on this simple comparison, we conclude that 4th Street is more resilient than Wurster Hall to the extreme air pollution event. This cumulative comparison is possible only after the air pollution event has occurred and is useful for ranking buildings across a large portfolio (e.g. campus, district, town) and identifying which buildings should have resource priority.

From the occupants' perspective, PM₂.₅ concentration and amount of time they were exposed play a critical role in health implications. In order to evaluate this in the context of building resilience, we must simultaneously evaluate ambient concentration and exposure time, which is done with the developed E-index tool. In Fig 4, the x-axis represents days during

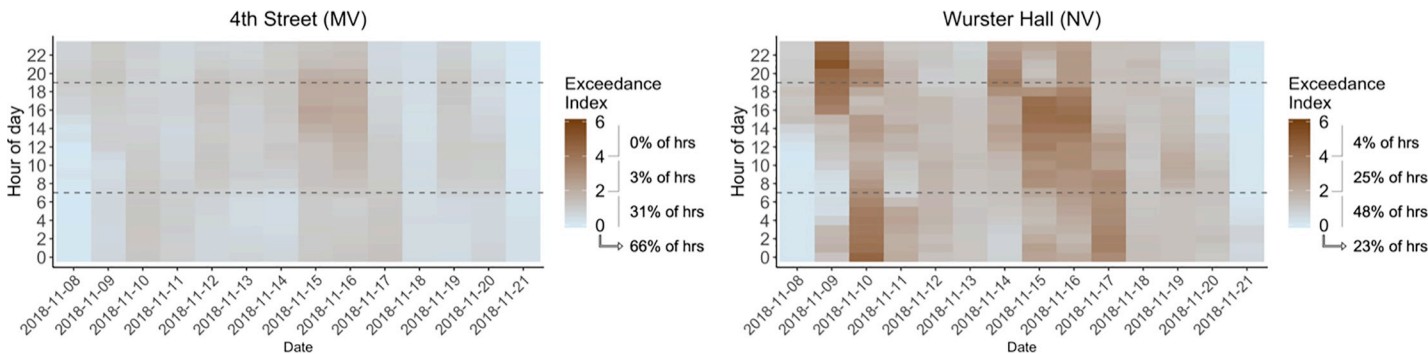

**Fig 4.** Exceedance index PM$_{2.5}$ heat map calculated for WHO 24 h exposure threshold: a) 4$^{th}$ Street and b) Wurster Hall.

the pollution episode, the y-axis represents hour of the day, with horizontal dotted lines indicating typical building operation hours of 07:00 to 19:00. For Fig 4 at the whole building scale, E-index is calculated using the median indoor PM$_{2.5}$ concentration of all indoor sensors for each hour; E-index can also be calculated at zone level, but we do not show these results. This representation visually shows exposure level and duration.

Results in Fig 4 show that 4$^{th}$ Street whole building E-index was below WHO suggested threshold for 66% of the hours (indicated as E-index of 0) and was one or greater for 34% of the hours, with only 3% being two or greater, meaning that only 3% of the pollution episode hours had median indoor PM$_{2.5}$ concentration more than or equal to twice the WHO threshold level. Overall average E-index, calculated by averaging hourly values for the entire pollution episode, for 4$^{th}$ Street was 0.82, suggesting that the building as a whole was resilient to outdoor air pollution during this episode, and that the tight building envelope paired with two-staged particle filtration of MERV 8 and MERV 13 at the air handler was effective at blocking PM$_{2.5}$ penetration and providing acceptable IAQ conditions.

Results for the Wurster Hall show that PM$_{2.5}$ concentration exceeded the recommended level for 77% of the hours. E-index was in the range from one to less than two for 48% of the hours, from two to less than four for 25% of the hours, and four or above for 3% of the hours. From an hourly analysis, we can see that Wurster Hall had periods that could potentially cause acute effects on occupants, especially for those with pulmonary health issues. One potential way to think about the role of interventions would be to eliminate periods where E-index was above a specified threshold (e.g. level that causes acute effects). For the entire pollution episode, overall average E-index was 1.69, suggesting that Wurster Hall requires intervention in order to be operational during an air pollution episode. Although overall average E-index enables us to compare and rank building resilience performance, the simplification to a single number representation omits important results: that for 29% of the hours Wurster Hall indoor concentration levels exceeded the WHO 24h-mean PM$_{2.5}$ exposure thresholds by more than two and was at a level that might cause acute health effects during occupied hours.

**3.2.2 Cumulative and instantaneous I/O ratio.** As described in [17], I/O ratio provides information about the quantity of outdoor particles that enter indoors when indoor sources can be neglected. Considering cumulative I/O ratio at the whole building scale, the median I/O ratio over the entire pollution episode was 0.27 for 4$^{th}$ Street and 0.67 for Wurster Hall. Cumulative whole building I/O ratio is a general indicator of building resilience and can aid in comparing amongst buildings, but it does not contain information on more granular spatial or temporal variations. In Fig 5, we show the cumulative hourly I/O ratio across all sensors and

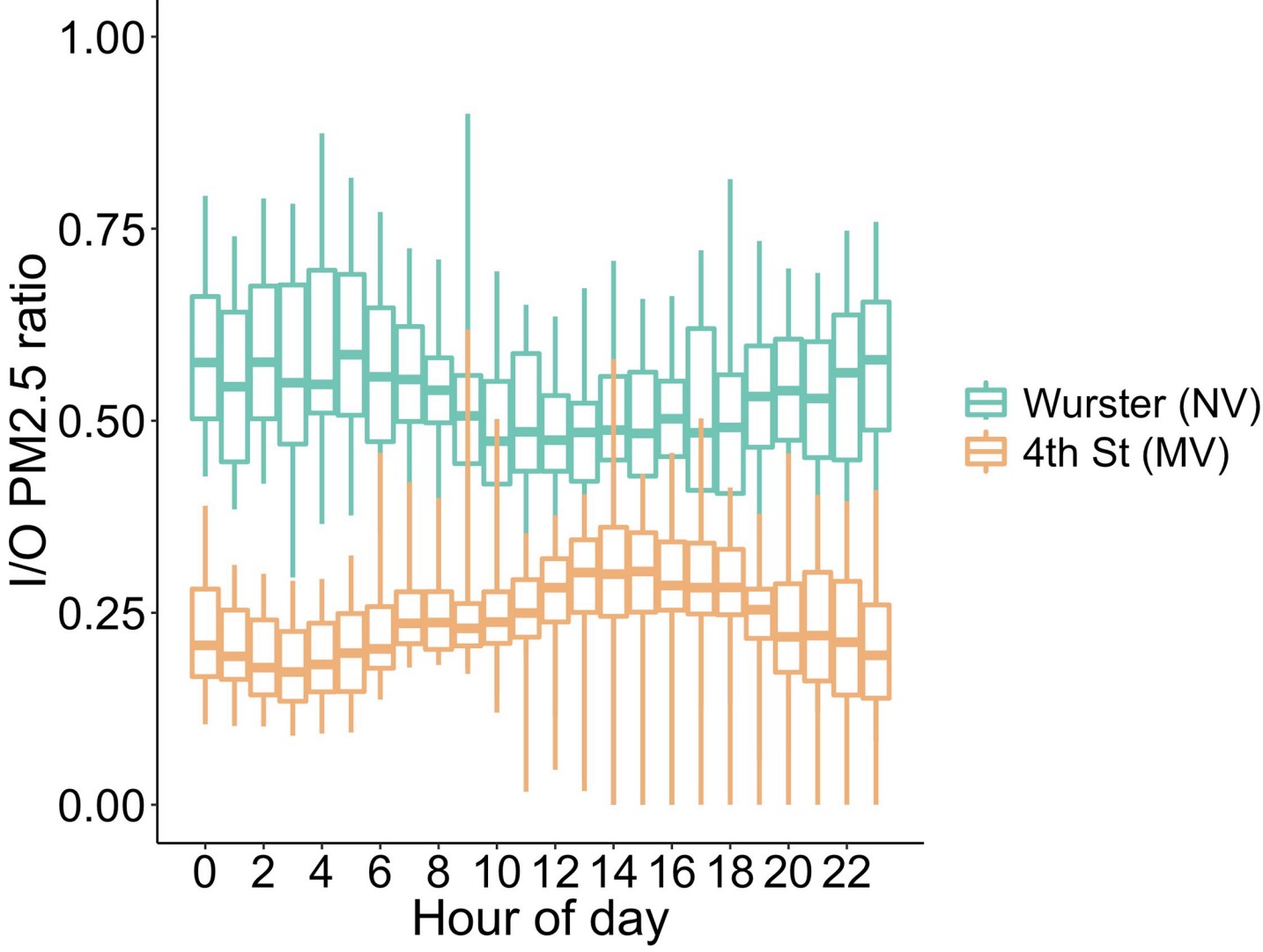

**Fig 5. Hourly distribution of I/O ratios for the whole duration of the air pollution episode caused by the Chico Camp fire (sliding temporal window).**

across the entire forest fire event. This can aid in identifying time dependent operational aspects at the whole building scale.

In some instances, the cumulative I/O ratio can provide insights about $PM_{2.5}$ penetration and ingress. As seen in Fig 5, there were trends in I/O ratio by hour of day during the entire pollution episode in each building. The 4th Street had a higher probability of a larger I/O ratio during the HVAC system operating hours compared to the non-operating hours. During this period, median I/O ratio was 0.33 during operating hours compared to 0.21 during non-operating hours. This indicated that $PM_{2.5}$ ingress is the dominant pathway of outdoor particle entry. The higher I/O ratios observed in Wurster Hall compared to 4th Street is due to the building being less airtight. Wurster Hall has operable windows designed for natural ventilation, and although the windows were closed (based on contact sensor readings) during the pollution event, the $CO_2$ decay rate shows infiltration of 0.4 $h^{-1}$, as opposed to 0.1 $h^{-1}$ for 4th Street.

Cumulative I/O ratio can also be viewed on smaller spatial scales, such as individual rooms or zones, which allows for comparison and assessment within a building, similar to the

comparison between buildings. We can use cumulative I/O ratio as a tool to assess (post-event) the effectiveness of an intervention, such as changes in building control strategy, economizer regime, air handler filter grade, or changes in recirculated-to-outdoor air ratio. Local level of intervention might improve conditions in a particular room or zone without necessity for larger scale intervention or any significant effect on a building scale.

Cumulative I/O ratio describes overall resilience to an air pollution episode but is limited in that it is only a post-event evaluation tool. IoT sensing can also be used to show instantaneous I/O ratio that can aid in real-time evaluation, such as quickly assessing whether a portable filter is appropriately sized for a room. For example, at the room scale, the instantaneous (e.g., hourly) I/O ratio results in Figs 6A and 7B for Wurster Hall show that different indoor locations can vary from the median building I/O ratio (i.e., the median of all indoor sensors). Fig 6A depicts the effectiveness of a portable filter intervention in room 348 on Nov. 15th. When in use, the portable filter quickly reduced the I/O ratio by more than 50%. The portable filter was manually turned on when the occupant entered the room then off when they left and that was repeated one more time during the day. This results in the sharp decreases and increases in I/O ratio in the middle of the day. Fig 6B shows that room 232, which is adjacent to a hallway with an entrance door to the building, can have ~40% higher instantaneous I/O ratio compared to the whole building median. This is likely due to elevated infiltration of particles caused by opening the entrance doors and movement of people [26]. This space can be considered as a pollution hot zone in the building and interventions should be considered, especially considering that the space had five occupants. As described in the introduction, current guidelines only have qualitative suggestions about portable filters, and this tool enables quantification of effectiveness of that or any intervention.

Instantaneous I/O ratio can also be viewed at the whole building scale, as shown in Fig 7 which depicts the median I/O ratio from all indoor sensors at each hour for the entire period of the event. Results show that in each hour 4th Street building was more resilient to high outdoor $PM_{2.5}$ levels and allowed less $PM_{2.5}$ to enter the building compared to Wurster Hall. In this way, Instantaneous I/O ratio can be used to evaluate the effectiveness of an intervention at the whole building scale in real time, such as increasing the filter grade in air handling units, or adjusting the outdoor air supply rate.

**3.2.3 Spatial and temporal variability within the buildings.** Placement of indoor $PM_{2.5}$ sensors is important for the evaluation of the building resilience; however, there is limited guidance on sensor density and appropriate placement. RESET and WELL v2 [27] suggest a sensing densities of one sensor per 500 m$^2$ and one sensor per 325 m$^2$, respectively. Sensing

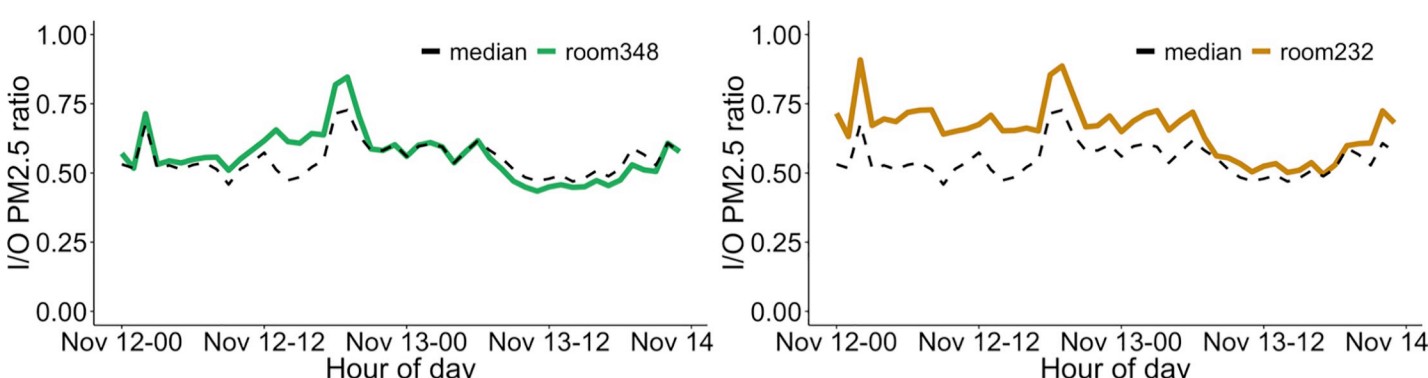

**Fig 6.** Instantaneous I/O ratios comparing median building operation and specific locations: a) operation of portable filter in room 348 and comparison to the building median I/O ratio; b) I/O ratio for the room 232 adjacent to the building exit door.

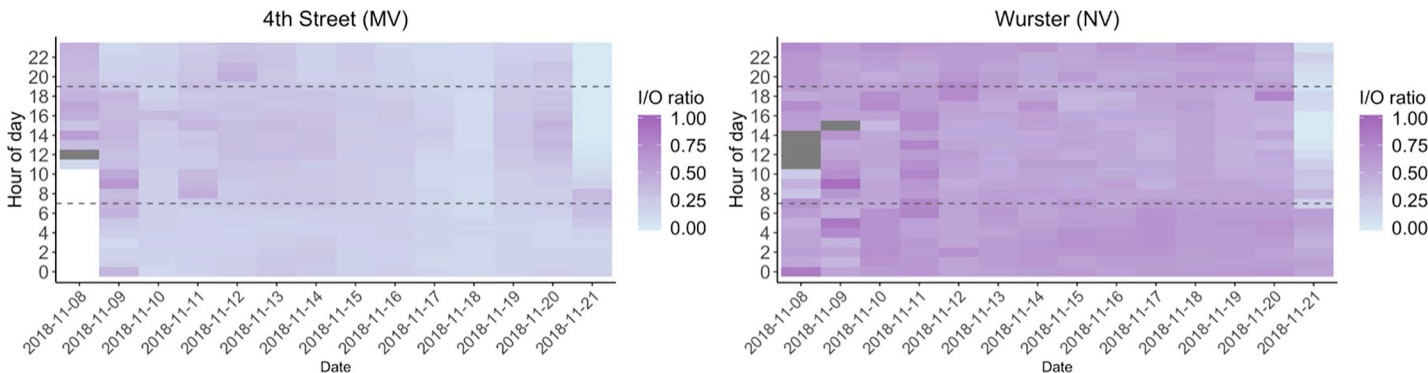

**Fig 7. Instantaneous I/O ratio for the entire building.** They are presented for the entire pollution episode with each point represents one hour period a) 4th Street; b) Wurster Hall.

density in the current study was two to five times higher. Results in Fig 8 show by how much measurements at each sensing location deviated from the hourly median calculated for all the sensors. The results show that the majority of deviations at each sensor location in both buildings were within ±10% (the measurement uncertainty), as indicated by the red dashed lines. Most of the sensors in 4th Street were placed in the open plan area, and one sensor was in a private office. Most of the sensors in Wurster Hall were in private offices. In both buildings, the results suggest homogenous PM$_{2.5}$ distribution under high outdoor pollution levels. Although results show a homogenous environment, we should not draw any general conclusion about the number of sensors necessary to properly characterize the indoor environment for other buildings. A larger sample of buildings would be necessary to determine sensor density.

### 3.3 Survey of occupants

From the occupant perspective, resilience can be assessed based on how disruptive an air pollution event was to "business as usual" operation. We assessed this using a survey to

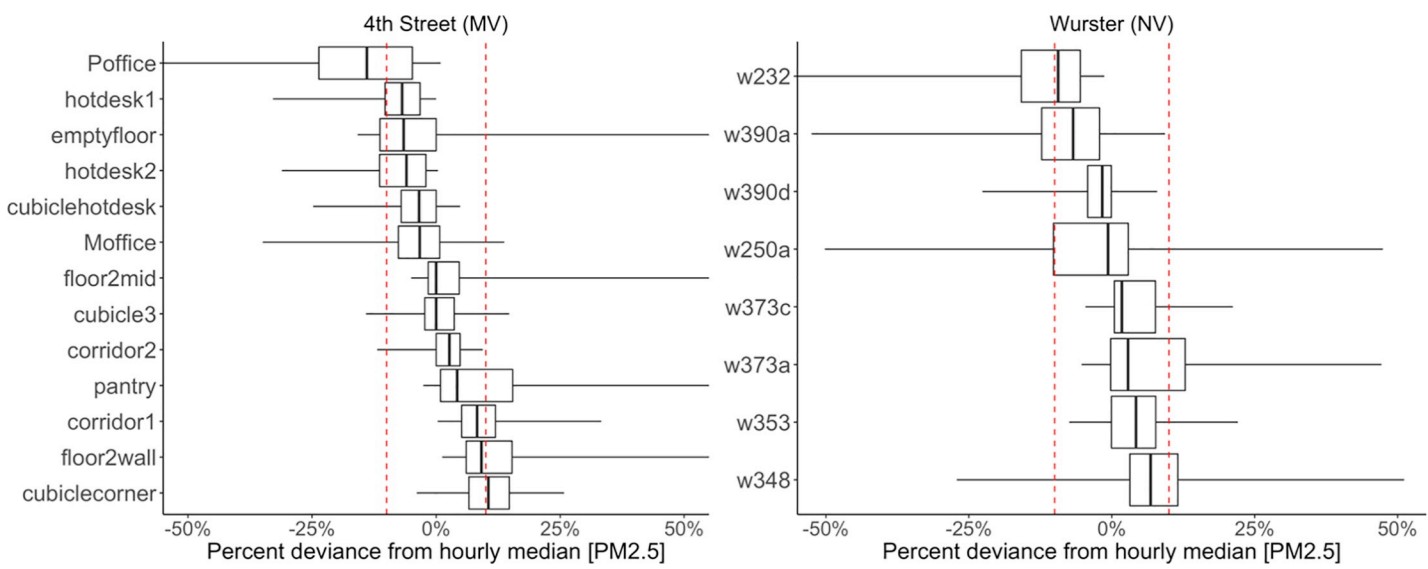

**Fig 8. Spatial variability of PM$_{2.5}$ levels represented as percent deviations from the mean level calculated with all the deployed sensors for each building.** The red dashed line represents the sensor measurement uncertainty of ±10%: a) 4th Street; b) Wurster Hall.

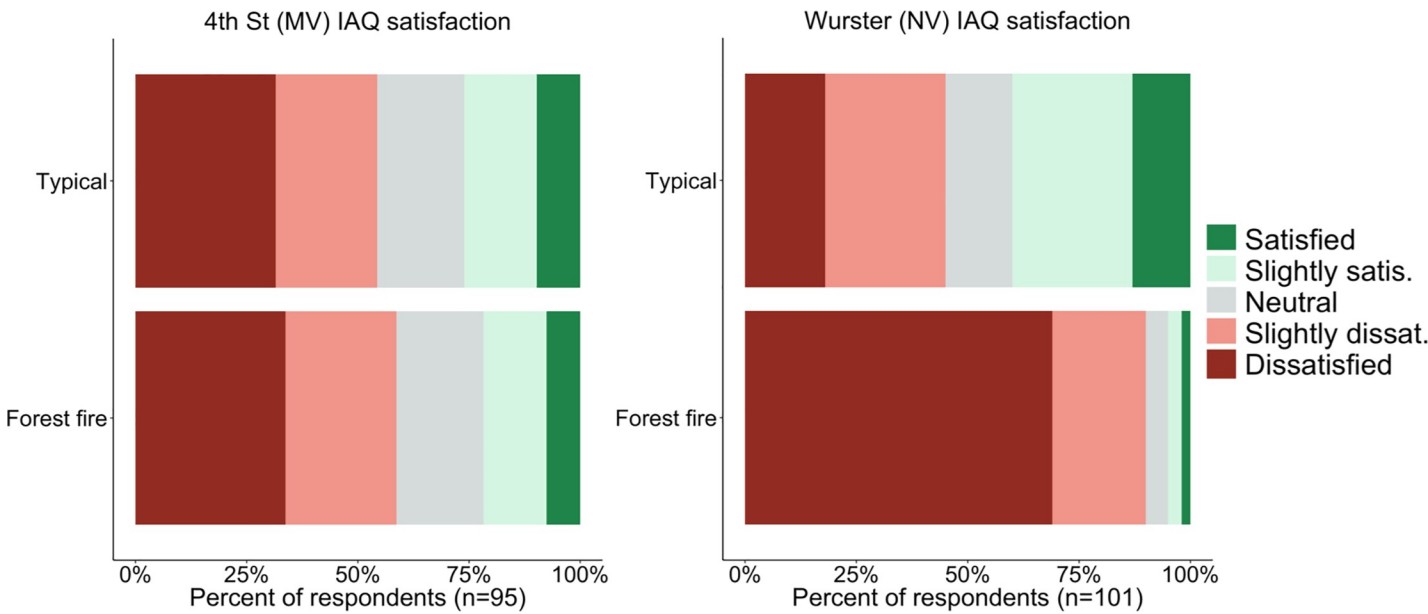

**Fig 9.** Occupant satisfaction with indoor air quality during typical conditions and during the Chico Camp fire in the: a) 4[th] Street (mechanically ventilated) and b) Wurster Hall (naturally ventilated).

understand occupants' air quality perception, self-reported productivity, and behaviour. Survey questions can be found in Supporting Information.

We received 95 complete responses in 4[th] Street (MV) and 101 in Wurster Hall (NV) from full-time occupants, correlating to response rates of 37% and 34%, respectively. In 4[th] Street, 78% have worked in the space for more than 1 year, and 89% of respondents work more than 30 hours per week. The demographics of Wurster Hall respondents were similar: 60% have worked in the space for more than 1 year, with the next largest group being those that have worked between 7 to 12 months. 66% of respondents work more than 20 hours per week in the building (45% of respondents work more than 30 hours per week). There was a fairly even distribution across age groups between 21 and over 50 in both buildings. It should be noted that there was minimal campus-wide or building-specific guidance on indoor pollution levels, strategies occupants could take in their workplace, or alternative places to work.

**3.3.1 Satisfaction with indoor air quality (IAQ).** Fig 9 shows the results of occupant satisfaction with indoor air quality during typical conditions and during the Chico Camp fire in the two building types. Under typical conditions, occupant satisfaction with IAQ was relatively low in both buildings, with 25% and 41% voting 'slightly satisfied' or 'satisfied' in 4[th] Street and Wurster Hall, respectively. There was a statistically significant difference in mean satisfaction scores (p<0.05), with Wurster Hall having higher satisfaction; however, the effect size was small to negligible (d = 0.3). There were 54% and 45% of respondents voting 'slightly dissatisfied' or 'dissatisfied' with IAQ in 4[th] Street and Wurster Hall, respectively. The lower satisfaction in 4[th] Street can be explained by the building's proximity to a cement facility, and, according to the building manager, occupants often complain of poor IAQ as a result even though there is a mechanical ventilation system.

Previous studies have found that humans are not accurate sensors of small changes in air pollutant concentrations when exposed to typical levels (i.e., typical to their experience), and that humans associate air quality with aspects of thermal comfort [28–30]. However, the survey responses indicate that during the extreme air pollution event, occupant perception aligned

with measured $PM_{2.5}$ concentrations in the two study buildings. In Wurster Hall, dissatisfaction with IAQ during the forest fire (i.e., votes for 'slightly dissatisfied' and 'dissatisfied') increased from 45% to 89% of respondents, with a statistically significant difference in mean satisfaction vote ($p < 0.05$) and a moderate to large effect size ($d = 1.3$) compared to typical conditions. Remarkably, there was only a slight increase in IAQ dissatisfaction from 54% to 57% in 4th Street, and the difference in mean satisfaction vote was not statistically significant ($p = 0.48$) compared to typical conditions.

The larger increase in dissatisfaction in Wurster Hall than 4th Street aligns with the buildings' low resilience, as seen in Fig 3 indoor $PM_{2.5}$ concentrations and Fig 7 $PM_{2.5}$ exceedance levels, which show that 4th Street was able to maintain indoor $PM_{2.5}$ concentrations below WHO threshold level most of the time. Notably, under the extreme pollutant conditions during the Chico Camp fire, occupants' perception of IAQ was in line with measured $PM_{2.5}$ concentration.

A confounding factor that we could not control is the air temperature in the buildings, which can impact the IAQ perception. Along with mechanical ventilation, 4th Street has air conditioning while Wurster Hall relies on operable windows for cooling. Outdoor temperatures were moderate at the time, with daily temperature highs between 15 to 22˚C. From the survey responses, air temperatures were only an issue in Wurster Hall classrooms with high occupancies. Three respondents mentioned that they opened windows or saw windows opened because classrooms became stuffy and hot, and they were forced to make a tradeoff with thermal comfort. Other locations such as private offices where the majority of full-time occupants reside were not likely affected by the air temperatures.

**3.3.2 Perceived impact to work productivity.** Poor air quality during the Chico Camp fire may have affected productivity by disrupting work schedules or routines, and by causing mental and physical stress on occupants. There is limited research linking wildfire pollutants, including $PM_{2.5}$ and carbon monoxide, to decreased cognitive abilities. We asked a series of questions to assess how occupants felt IAQ in their workplace affected productivity, including cognitive abilities and psychological state. Responses can help to assess productivity loss during the extreme pollution event and provide information for an intervention cost-benefit analysis [31,32].

Fig 10 shows the self-reported impacts of workplace IAQ on occupants' ability to get their work done. Similar to occupants' satisfaction with IAQ in Fig 9, occupants in both buildings reported similar levels of impact under typical conditions, with slightly higher reports of interference in 4th Street. Responses for the Chico Camp fire period are in line with previously reported trends of Wurster Hall occupants reporting more severe impact. Both buildings have statistically significant ($p < 0.05$) difference in mean response between typical conditions and during the Chico Camp fire, but the percent of respondents that reported that workplace IAQ interferes (i.e., considering 'somewhat interferes' and 'interferes') with productivity increased more in Wurster Hall (38% to 81%) than in 4th Street (45% to 52%) when comparing typical conditions to the Chico Camp fire.

In addition to asking directly about productivity, we wanted to understand how occupants felt (i.e., their psychological state) in regards to IAQ once they entered their building from the outside where there was high pollutant concentration, as seen in Figs 3 and 4. Although personal exposure varied by mode of transportation (e.g., by bike, by bus, walking, or by car) and use of N95 face masks, all occupants had some level of exposure and were likely aware of the highly publicized outdoor air quality. The results, shown in Fig 11, have a significant difference ($p < 0.05$) in mean relief vote between the buildings with a small effect size ($d = 0.42$). More respondents felt relieved in 4th Street and more felt not relieved in Wurster Hall, which aligns with the overall trend of measured performance for those buildings. Although the $PM_{2.5}$

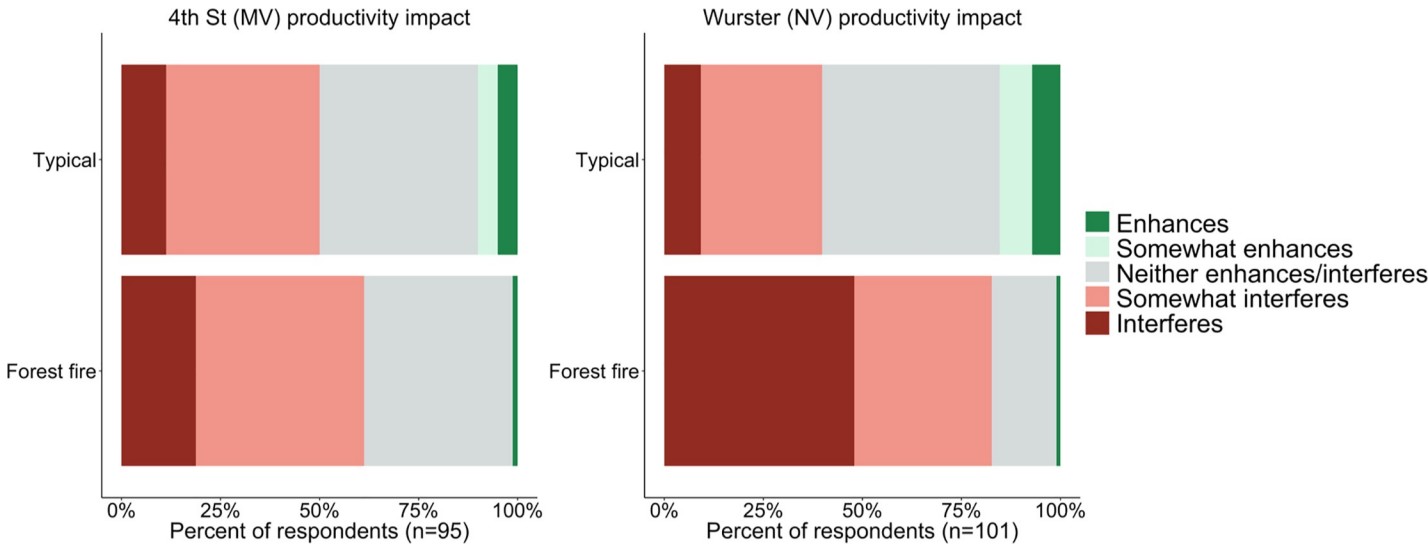

**Fig 10.** Occupant self-reported impact of IAQ on their productivity and ability to get their work done a) 4th Street (mechanically ventilated) and b) Wurster Hall (naturally ventilated).

concentrations in 4th Street were below WHO recommended level the majority of the time, 72% of respondents reported being 'not relieved' or only 'slightly relieved'. This could be due to indoor $PM_{2.5}$ concentrations being higher than occupants are typically exposed to, unsatisfactory IAQ during typical conditions, or lack of knowledge that indoor pollution was below the risk levels for healthy adults.

When survey results from Figs 10, 11 and 12 are combined with IoT $PM_{2.5}$ measurements in Figs 4 and 5, we see a trend that shows occupants perceived lower $PM_{2.5}$ in 4th Street compared to Wurster Hall. This provides evidence that surveys can be a useful tool in intervention assessment and aid in intervention decision-making during the severe outdoor air pollution events.

**3.3.3 Occupant behavior and strategies to reduce exposure.** Another approach to understand occupants' perception of IAQ is by reviewing their actions to reduce exposure to pollutants during the Chico Camp fire. Occupants identified which building features they controlled and what strategies or devices, if any, they used to reduce exposure to pollutants while in their workplace during the Chico Camp fire. The results in Fig 12 shows that 24% in 4th Street and 29% in Wurster Hall wore a face mask in their workplace at some point during the fire. A respondent from Wurster Hall stated, "During the forest fires, smoke filled the room to the point that I was wearing my mask inside." N95 face masks can reduce individual exposure, but are still a controversial issue especially related to effectiveness of non-fitted masks [13] and recommendations for some groups (e.g., pregnant women, small children, people with respiratory challenges) not to wear them. Additionally, we can reasonably expect that wearing face masks affects productivity by obstructing normal operation within the building, reducing interaction with colleagues, degrading general comfort, and increasing $CO_2$ inhalation which has been linked to cognitive function [33].

While occupants' direct responses about satisfaction and perceived impacts in each building align with IoT $PM_{2.5}$ sensing measurements, their reported behavior for wearing face masks did not. Despite indoor $PM_{2.5}$ concentration being typically below the WHO threshold level in 4th Street and occupants generally reporting higher satisfaction with and lower impact from IAQ during the Chico Camp fire, occupants still wore masks indoors at about the same rate as

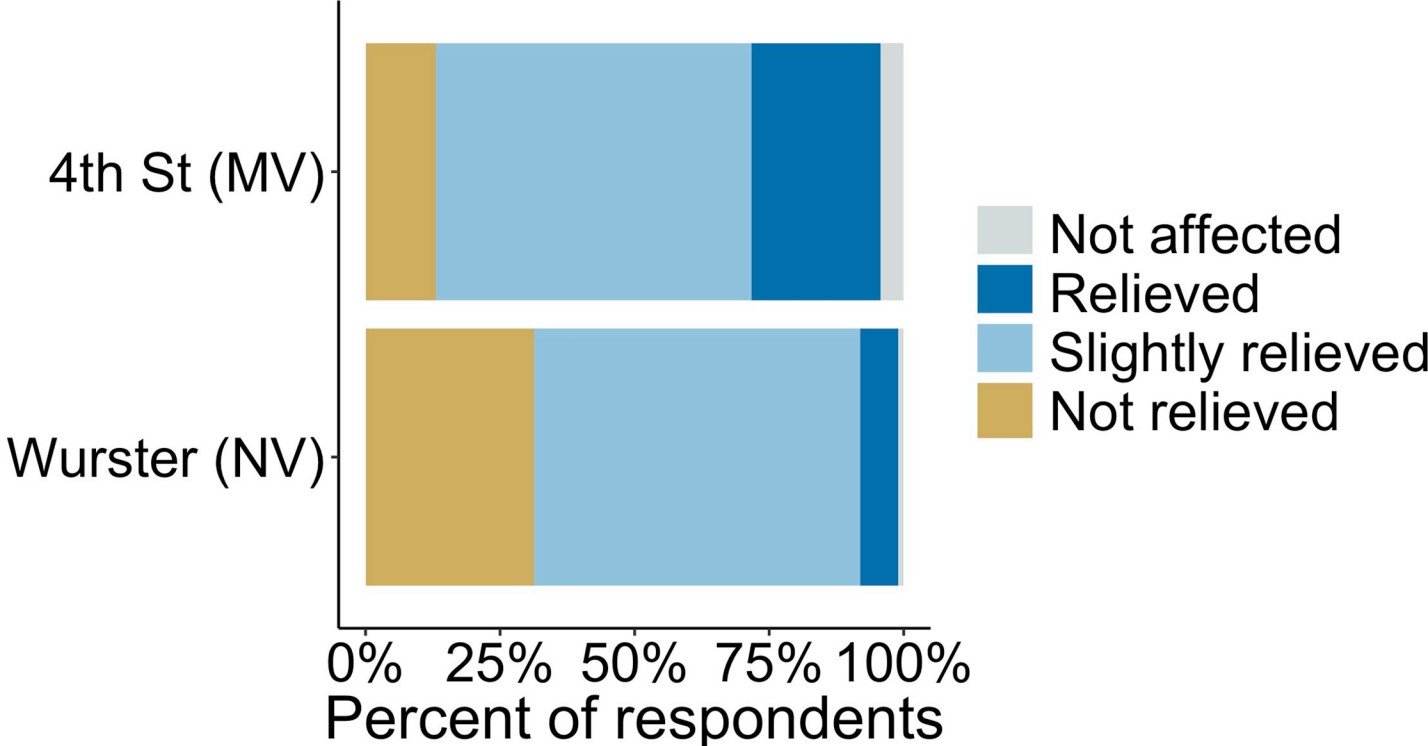

**Fig 11. Occupant responses to "When you entered your workspace from the outdoors did you feel relief regarding the air quality?" during the Chico Camp fire.** The question assumes that respondents were negatively impacted by outdoor conditions; we included a response for "I was not affected by the outdoor conditions", indicated as "not affected".

those in Wurster Hall. The use of face masks in both buildings may have been driven by concern over visual appearance of outdoor air conditions (e.g., grey fog), publicized air quality index (AQI) for outside air, lack of knowledge about air quality in the workplace, and lack of alternative solutions.

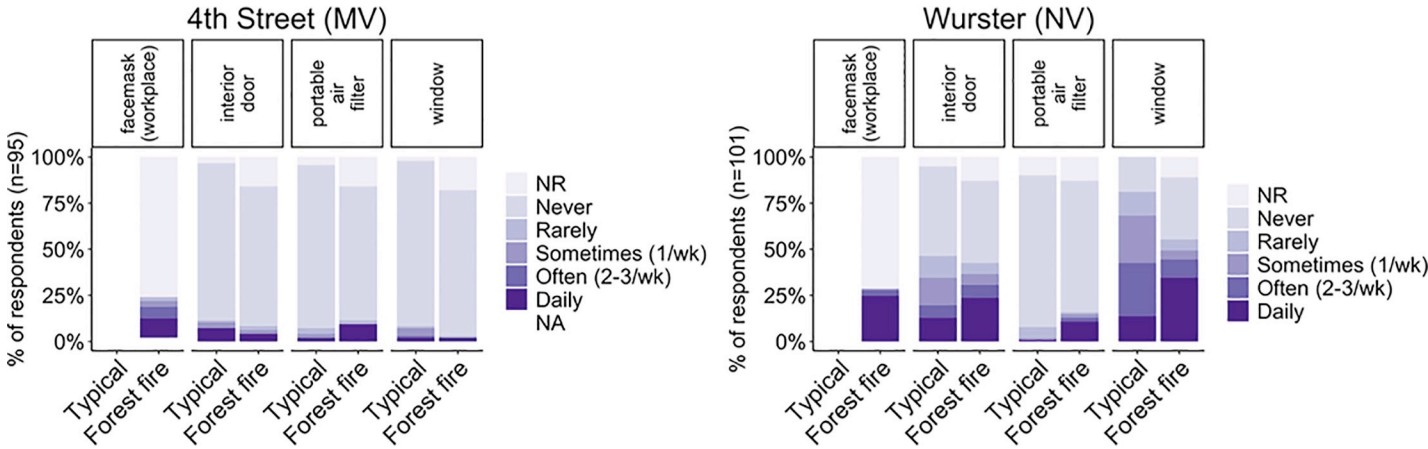

**Fig 12. Personal and building features that occupants reported using in their workplace: a) 4[th] Street (mechanically ventilated) and b) Wurster Hall (naturally ventilated).** The option for a face mask was only listed in the forest fire section, and a branching question allowed respondents to indicate if they wore a face mask in their workplace or outdoors. NR is no response given.

Other strategies to avoid pollutant exposure include operating interior doors and windows to the outside in Wurster Hall. As shown in Fig 12, 4th Street occupants did not report high use of any controllable features, mostly because the space is largely open office (i.e., few interior doors) with a limited number of operable windows in select locations. The use of portable air filters was low in both building, with 11% using this device in 4th Street and 16% using in Wurster Hall, less than the number of occupants wearing face masks. We did not collect information on the air filter type or size, but as seen in Fig 6A, a portable air filter can improve IAQ in these events. Some of the possible reasons that portable filters were not more widely used are upfront costs and lack of knowledge or means to measure effectiveness of air filters, which must be properly sized for room volume.

**3.3.4 Survey as a building performance assessment tool.** Comparing the IoT $PM_{2.5}$ measurements with the occupant IAQ satisfaction results shows that a survey has the potential to be an effective indicator of general IAQ during the high outdoor air pollution events; however, the survey only provides qualitative information and can assess whether a building is able to maintain "business-as-usual" for occupants. As in the case of Wurster Hall (NV), the survey indicated low IAQ satisfaction, but sensor measurements were necessary to quantify the indoor pollutant concentrations and mechanisms for air pollutant entry into the building. As opposed to the IoT devices, a survey is more often used as a retrospective tool (i.e., can only be applied after the air pollution event occurs), and it could be used to assess the resiliency of buildings or effectiveness of interventions to maintain "business-as-usual".

As indicated by respondents wearing face masks indoors, maintaining $PM_{2.5}$ concentration levels within the prescribed limits, either through building level features such as air handler's filter grade or alternatives solutions like portable air filters, might not be enough to prevent occupants from feeling at risk. A key missing piece was ensuring that occupants are aware of the conditions in the space. Results from this survey only point to this possibility, and need to be validated. Without sensor equipment from this study, neither 4th Street nor Wurster Hall had mechanisms to monitor indoor $PM_{2.5}$ levels and thus, there was a limited ability to provide building-specific information. This identifies a gap and an additional aspect to consider for a survey to probe into the effectiveness of communicating air quality data.

# 4. Recommendations for building resilience evaluation

The general trending direction of IoT environmental sensing application and research is to create urban air pollution maps and increase knowledge about the pollution distribution on a city scale; however, this maintains the assumption that outdoor measurement is a good proxy for personal exposure of the entire population within the vicinity of the measurement location. This study shows new and additional value of dense urban environmental IoT sensing networks and addresses the missing gap in integration between urban and building scale IoT sensing, especially considering that humans in developed parts of the world spend approximately 90% of their time indoors [34].

The use of these tools is widespread and can serve several purposes including benchmarking building performance, assessing the effectiveness of interventions, and understanding occupant perception and behavior. Benchmarking building resilience is useful for an entity that manages a large building portfolio (e.g. university campus consisting of ~100 buildings). E-index and I/O ratio are benchmarking tools that can rank and prioritize buildings for improvements and inform occupants of safety. The use and potential applications of these tools at various temporal and spatial scales are listed in Table 2. E-index provides information on exposure level compared to a specified threshold (e.g., WHO) and can identify "safe" buildings or spaces within buildings; similarly, it will identify buildings with high air pollutant

**Table 2. Summary of tools proposed for building resilience evaluation.**

| | E-index | | I/O ratio | |
| --- | --- | --- | --- | --- |
| | Instantaneous | Cumulative (entire episode) | Instantaneous | Cumulative (entire episode) |
| **Whole building** | Monitor and inform building occupants of air pollutant exposure | Rank or benchmark buildings by occupant exposure relative to guideline exposure limits | Assess whole building interventions (e.g. AHU filter upgrades, changing controls, adjusting outdoor air intake or economizers) | Rank or benchmark buildings by ability to prevent infiltration/ penetration of outdoor pollutants. Predict building's performance under various outdoor air pollution scenarios |
| **By space** | Monitor and inform occupants of air pollutant exposure | Identify areas in a building with high potential occupant exposure | Assess zonal or room interventions (e.g. window operation, portable air filters, filter upgrades at the zone) | Rank or benchmark spaces |

exposure or hotspots within buildings. I/O ratio can quickly assess the effectiveness of potential interventions on a whole building or space-by-space scale, such as seen in Fig 6 with the portable air filter. It can also detect pollution hotspots, and predict what indoor pollution would be under different outdoor pollution scenarios. This is important during an extreme pollution event when actions need to be taken immediately. To illustrate the use of exceedance concept, comparison of 4th Street and Wurster Hall for 3 month period excluding forest fire episode is presented in the S3 Fig in Supporting Information. Beyond extreme air pollution events, local and daily pollution sources from vehicle emissions or emissions from factories represent other examples where the IoT sensing tools can help identify issues and characterize building resilience.

Creating a database with E-index and/or I/O ratio for a large and diverse set of buildings (i.e., with various typologies, ventilation air delivery strategies, cooling and heating systems, control sequences, facade air tightness, or pressure differentials between indoor and outdoors) will aid in identifying safe buildings and effective practices that can be implemented when extreme air pollution events occur. Together with low-cost IoT sensing, these tools can map out the indoor air pollution risk in a city or campus to inform organizational policies during an extreme event. This could lead to incorporating requirements for building resilience against wildfire-associated air pollution into the design stage of buildings. It can also lead to the development of clean air safe shelters for those at risk, such as homeless people, those who were evacuated from the wildfires affected areas, those with respiratory problems, or those who cannot afford portable filters.

Using the suggested criteria for a resilient building during extreme events, which is to maintain "business as usual" occupant behavior and perception, buildings can be considered successful if occupants feel safe enough indoors not to wear face masks. It is necessary to understand how to achieve this, and one component in this process is collecting IAQ information which can be achieved with IoT sensing. In future work, the survey can be used as an assessment tool and can be reduced to the questions necessary to understand perception and behavior. Additionally, it should include aspects about the effectiveness of methods to increase occupant awareness of indoor air quality during pollution episodes. Further, if IoT sensing measurements are not available or feasible for a building, the study results indicate that occupant perception could be used as a proxy for IAQ; although additional studies in various buildings should be conducted to confirm this.

In interpreting the study results, a few limitations should be recognized. Firstly, sample of only two buildings and a relatively limited number of responses were used for the analysis. As such, the I/O ratio, E-index and occupancy response results reported here cannot be indicative of all buildings and all people. Secondly, the assumption that one local measurement location

(placing the sensor on the roof of each of the buildings) is sufficient to represent the outdoor pollution level for the entire building and conclude how the building operates in respect to the penetration of outdoor pollution indoors. Another limitation is a restricted utility of I/O ratio for recommending the specific building interventions, unless used for building resiliency evaluation against the elevated outdoor air pollution. However, the tools can be effective in dry areas prone to wildfires and in heavily polluted cities around the world. Notwithstanding the limitations, our study has not only provided useful preliminary data on which further research, but also offered a methodological contribution towards the assessment of building protection from air pollution of outdoor origin elsewhere.

## 5. Conclusions

The tools we used in this study are methods to evaluate building resilience to extreme outdoor air pollution episodes using IoT sensing indoors and outdoors, coupled with survey-based information of occupant perception and behaviour. We demonstrated the application of the tools on two buildings with different modes of ventilation. We characterized the resilience of the buildings on different temporal and spatial scales using the well-established I/O ratio and a newly proposed E-index. These tools can be used for a single building or across a portfolio of buildings for resilience ranking. Meanwhile, the survey tool provides qualitative insights on the connection between air pollution levels and occupants' perception and behavior, which allows assessment of a building's ability to be resilient, meaning that occupants can maintain "business as usual" view.

Results indicate that even under the extreme outdoor air pollution events, outdoor $PM_{2.5}$ concentrations can be significantly different across a region, and therefore, outdoor sensors need to be localized to properly evaluate building resilience. Using the tools developed in this study comparing the resilience of mechanically ventilated 4[th] Street building and naturally ventilated Wurster Hall building indicates:

- Indoor $PM_{2.5}$ concentration over the entire Chico Camp Fire event was 21 $\mu g/m^3$ for 4[th] Street and 36 $\mu g/m^3$ for Wurster Hall. The cumulative median I/O ratio over the entire pollution episode was 0.27 for 4[th] Street and 0.67 for Wurster Hall. Overall E-index for 4[th] Street was 0.82, suggesting that the whole building was resilient to air pollution while overall E-index was 1.69 for Wurster Hall suggesting that interventions are necessary in order to be operational during an air pollution episode.

- The survey revealed that occupant perception of workplace IAQ aligns with measured $PM_{2.5}$ in the two buildings, with greater self-reported impacts in Wurster Hall which had significantly worse IAQ than 4th Street. After entering indoor environment coming from outside, occupants in Wurster Hall felt significantly less relieved ($p<0.05$) than those in 4th Street As many as 31% of occupants that did not feel relieved in Wurster Hall and only 13% in 4th Street Building. The results also revealed that a large portion of occupants wore face masks in both buildings, even though the $PM_{2.5}$ concentration was below the WHO threshold level. In 4th Street, which was likely driven by lack of alternative solutions and knowledge of PM2.5 levels. This suggests that occupants did not feel or did not know that the indoor air was at the acceptable level. We use this behaviour to suggest criteria for building resiliency.

## Supporting information

**S1 Fig.** Distribution of sensors in the 4th Street Building a) floor 2; b) floor 3. (TIFF)

**S2 Fig.** Distribution of sensors in the Wurster Hall a) floor 2; b) floor 3.
(TIFF)

**S3 Fig. E-index for 4th Street and Wurster Hall from November 11th until January 28th.**
(TIFF)

**S1 Data. PM2.5 measurments of all sensors from 10/8/2018 until 10/21/2018.**
(CSV)

## Acknowledgments

We would like to thank Professor Mark Modera from University of California at Davis and Professor Wolfgang Rogge from University of California at Merced for their collaboration on this project.

## Author Contributions

**Conceptualization:** Jovan Pantelic, Megan Dawe, Dusan Licina.

**Data curation:** Jovan Pantelic, Megan Dawe.

**Formal analysis:** Jovan Pantelic, Megan Dawe.

**Funding acquisition:** Jovan Pantelic.

**Investigation:** Jovan Pantelic, Megan Dawe.

**Methodology:** Jovan Pantelic, Megan Dawe.

**Project administration:** Jovan Pantelic.

**Resources:** Jovan Pantelic.

**Supervision:** Jovan Pantelic.

**Writing – original draft:** Jovan Pantelic, Megan Dawe.

**Writing – review & editing:** Dusan Licina.

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
