## [Decision Letter · Decision Letter 0]

23 Aug 2019

PONE-D-19-21370

Use of IoT sensing and occupant surveys for determining the resilience of buildings to forest fire generated PM2.5

PLOS ONE

Dear Dr Pantelic,

Thank you for submitting your manuscript to PLOS ONE. After careful consideration, we feel that it has merit but does not fully meet PLOS ONE’s publication criteria as it currently stands. Therefore, we invite you to submit a revised version of the manuscript that addresses the points raised during the review process.

ACADEMIC EDITOR: The references need to check properly to follow the guide for author.

We would appreciate receiving your revised manuscript by Oct 07 2019 11:59PM. To enhance the reproducibility of your results, we recommend that if applicable you deposit your laboratory protocols in protocols.io, where a protocol can be assigned its own identifier (DOI) such that it can be cited independently in the future. For instructions see: http://journals.plos.org/plosone/s/submission-guidelines#loc-laboratory-protocols

We look forward to receiving your revised manuscript.

Kind regards,

Bawadi Abdullah

Academic Editor

PLOS ONE

Journal Requirements:

2. We note that Figure 2 in your submission contain [map/satellite] images which may be copyrighted. All PLOS content is published under the Creative Commons Attribution License (CC BY 4.0), which means that the manuscript, images, and Supporting Information files will be freely available online, and any third party is permitted to access, download, copy, distribute, and use these materials in any way, even commercially, with proper attribution. For these reasons, we cannot publish previously copyrighted maps or satellite images created using proprietary data, such as Google software (Google Maps, Street View, and Earth). For more information, see our copyright guidelines: http://journals.plos.org/plosone/s/licenses-and-copyright.

You may seek permission from the original copyright holder of Figure(s) [#] to publish the content specifically under the CC BY 4.0 license. 

If you are unable to obtain permission from the original copyright holder to publish these figures under the CC BY 4.0 license or if the copyright holder’s requirements are incompatible with the CC BY 4.0 license, please either i) remove the figure or ii) supply a replacement figure that complies with the CC BY 4.0 license. Please check copyright information on all replacement figures and update the figure caption with source information. If applicable, please specify in the figure caption text when a figure is similar but not identical to the original image and is therefore for illustrative purposes only.

4. Please amend your current ethics statement to confirm that your named institutional review board or ethics committee specifically approved this study.

'The authors would like to acknowledge CITRIS at University of California Berkeley for sponsoring this research with the grant number 69085-24003-44-NHJVN.

This research was also partially supported by the Republic of Singapore's National Research Foundation through a grant to the Berkeley Education Alliance for Research in Singapore (BEARS) for the Singapore-Berkeley Building Efficiency and Sustainability in the Tropics (SinBerBEST) Program.'

'PI-JP; Co-PI MM; Co-PI WR

The authors would like to acknowledge CITRIS and the BANATAO Institute at

University of California Berkeley for sponsoring this research with the grant number

69085-24003-44-NHJVN.

Sponsor was not involved in the study or manuscript preparation.'

6. Thank you for stating the following in your Competing Interests section: 

'No'

Additional Editor Comments (if provided):

Reviewers' comments:

Reviewer's Responses to Questions

**Comments to the Author**

1. Is the manuscript technically sound, and do the data support the conclusions?

Reviewer #1: Yes

Reviewer #2: Yes

2. Has the statistical analysis been performed appropriately and rigorously? 

Reviewer #1: Yes

Reviewer #2: Yes

3. Have the authors made all data underlying the findings in their manuscript fully available?

Reviewer #1: No

Reviewer #2: Yes

4. Is the manuscript presented in an intelligible fashion and written in standard English?

Reviewer #1: Yes

Reviewer #2: Yes

5. Review Comments to the Author

Reviewer #1: In general the content of the paper is written in a good flow. The Results and conclusions meet the objectives stated in the Introduction. The methodology is strengthen by benchmarking with the standards used in the actual operation.

Reviewer #2: GENERAL REPORT

1.1 General Comments

The paper is well written, with clear objectives. The topic is an important subject. However, I have the following comments for revision consideration to make it more appeal to the general audience of Plos One.

1.2 Flow of the Paper

The author used logical connections between ideas and sentences. The transition of paragraphs is linked with strong sentences.

1.3 Strength of the Paper

• This study aiming at an examination of how a combination of an IoT environmental sensing network, implemented locally outdoors and inside the building. Further, the occupant survey has been utilized to understand and evaluate building resiliency to urban scale air pollution has not yet been attempted to my knowledge.

• A study with such objectives could fill some significant gaps in defining and quantifying building resiliency, the effectiveness of chosen interventions, and general understanding of occupant exposure.

1.4 Weakness of Paper

• Yet, while there are some merits of the study as mentioned above, this study needs some improvements. There is a need to highlight the implication of this study. The introduction section needs to be more concise in terms of elaborating on the research gap.

1.5 Significance of the Paper

The study is significant in defining and quantifying building resiliency by providing an understanding of the accuracy of IoT sensing equipment. The survey tool provides qualitative insights on the connection between air pollution levels and occupants’ perception and behavior, which allows assessment of a building’s ability to be resilient.

1.6 Clarity of the study

Clearly define focal constructs and terms. Moreover, the research has clarity in defining the objectives of the study and relating the results.

1.7 Terminology Used

1.8 Use of Language

The paper is well written. Language is easy to understand and explains the matter correctly. However, some sentences are too long, making the reader hard to understand. There are some typos need to be fixed (for example see 2.4.3 paragraph 2 line 7).

6. PLOS authors have the option to publish the peer review history of their article (what does this mean?). If published, this will include your full peer review and any attached files.

Reviewer #1: No

Reviewer #2: No

---

## [Author Response · Author response to Decision Letter 0]

11 Sep 2019

Journal Requirements:

1. We note that Figure 2 in your submission contain [map/satellite] images which may be copyrighted. All PLOS content is published under the Creative Commons Attribution License (CC BY 4.0), which means that the manuscript, images, and Supporting Information files will be freely available online, and any third party is permitted to access, download, copy, distribute, and use these materials in any way, even commercially, with proper attribution. For these reasons, we cannot publish previously copyrighted maps or satellite images created using proprietary data, such as Google software (Google Maps, Street View, and Earth). For more information, see our copyright guidelines: http://journals.plos.org/plosone/s/licenses-and-copyright.

Answer: Figure 2 is removed from the manuscript as we didn’t see too much of its use.

Answer: Thank you for the useful suggestion - the full survey questionnaire is included in the Supporting Information document.

3. Please amend your current ethics statement to confirm that your named institutional review board or ethics committee specifically approved this study.

Answer: In section 2.3.4 sentence is included: The University of California at Berkeley Institutional Review Board approval was obtained Protocol ID:2019-02-11802.

4. Have the authors made all data underlying the findings in their manuscript fully available?

Reviewer #1: No

Answer: All the data from the study is now included.

General Comments

Reviewer #1: In general the content of the paper is written in a good flow. The Results and conclusions meet the objectives stated in the Introduction. The methodology is strengthened by benchmarking with the standards used in the actual operation.

Answer: Thank you for your comments.

Reviewer #2: 

1. Weakness of Paper

Yet, while there are some merits of the study as mentioned above, this study needs some improvements. There is a need to highlight the implication of this study. The introduction section needs to be more concise in terms of elaborating on the research gap.

Answer: The introduction section has been revised and is more concise now. The research gap is explained better. All the changes are kept as track changes in the manuscript.

2. Use of Language

The paper is well written. Language is easy to understand and explains the matter correctly. However, some sentences are too long, making the reader hard to understand. There are some typos need to be fixed (for example see 2.4.3 paragraph 2 line 7).

Answer: Long sentences have been identified and rewritten. You can see them as track changes in the manuscript.

---

## [Editor Report · Decision Letter 1]

16 Sep 2019

Use of IoT sensing and occupant surveys for determining the resilience of buildings to forest fire generated PM2.5

PONE-D-19-21370R1

Dear Dr. Pantelic,

We are pleased to inform you that your manuscript has been judged scientifically suitable for publication and will be formally accepted for publication once it complies with all outstanding technical requirements.

With kind regards,

Bawadi Abdullah

Academic Editor

PLOS ONE

Additional Editor Comments (optional):

The authors have complied with comments from reviewers.
---

## [Editor Report · Acceptance letter]

7 Oct 2019

PONE-D-19-21370R1 

Use of IoT sensing and occupant surveys for determining the resilience of buildings to forest fire generated PM_2.5_  

Dear Dr. Pantelic:

I am pleased to inform you that your manuscript has been deemed suitable for publication in PLOS ONE. Congratulations! Your manuscript is now with our production department. 

With kind regards,

on behalf of

Dr. Bawadi Abdullah 

Academic Editor

PLOS ONE